

# Projected Impact of Heat on Mortality and Labour Productivity under Climate Change in Switzerland

Zélie Stalhandske[1,2,*], Valentina Nesa[1,2,*], Marius Zumwald[1,2], Martina S Ragettli[3,4], Alina Galimshina[5], Niels Holthausen[6], Martin Röösli[3,4], and David N Bresch[1,2]

[1]ETH Zurich, Institute for Environmental Decisions, Zurich, 8092, Switzerland
[2]Federal Office of Meteorology and Climatology MeteoSwiss, Zurich-Airport, Zurich, 8058, Switzerland
[3]Swiss Tropical and Public Health Institute, Basel, 4051, Switzerland
[4]University of Basel, Basel, 4001, Switzerland
[5]ETH Zurich, Institute of Construction & Infrastructure Management, Zurich, Switzerland
[6]Amt für Abfall, Wasser, Energie und Luft, Canton of Zurich, Zurich, Switzerland
[*]these authors contributed equally to this work

**Correspondence:** Zélie Stalhandske (zelie.stalhandske@usys.ethz.ch)

**Abstract.** Extreme temperatures have reached unprecedented levels in many regions of the globe due to climate change and a further increase is expected. Besides other consequences, high temperatures increase the mortality risk and severely affect the labour productivity of workers. We perform a high-resolution spatial analysis to assess the impacts of heat on mortality and labour productivity in Switzerland and project their development under different Representative Concentration Pathway (RCP) scenarios, considering that no socio-economic changes takes place. The model is based on the risk framework of the Intergovernmental Panel on Climate Change (IPCC), which combines the three risk components: *Hazard*, *Exposure*, and *Vulnerability*. We model the two impact categories in the same spatially explicit framework and we integrate uncertainties into the analysis by a Monte Carlo simulation. We model, that first, about 670 people die today per year because of heat in Switzerland. Second, the economic costs caused by losses in labour productivity amount to around CHF 413 million (approx. $ 465 million) per year. Should we remain on an RCP8.5 emissions pathway, these values may double (for mortality) or even triple (for labour productivity) by the end of the century. Under an RCP2.6 scenario impacts are expected to slightly increase and peak around mid-century, when climate is assumed to stop warming. Even though uncertainties in the model are large, the underlying trend in impacts is unequivocal. The results of the study are valuable information for political discussions and allow for a better understanding of the cost of inaction.

## 1 Introduction

Temperatures around the globe keep increasing, with an observed mean positive warming of about one degree Celsius since pre-industrial levels (IPCC, 2013). As a result, extreme heat events are becoming more intense, last longer, and happen more frequently (Alexander et al., 2006; Coumou et al., 2013; Della-Marta et al., 2007; Fischer and Schär, 2009; Schär et al., 2004). These trends are particularly strong in central and southeastern Europe (Fischer and Knutti, 2014; Donat et al., 2013; Fischer


and Schär, 2010; Morabito et al., 2017). As extreme heat events become more common, humans experience an increase in environmental heat stress exposure (Willett and Sherwood, 2012; Zhao et al., 2015).

The impacts of heat on human metabolism have societal and economic consequences (Carleton and Hsiang, 2016; Haines et al., 2006; Mora et al., 2017; Kjellstrom et al., 2018). Heat-related illnesses happen when the human thermoregulatory capacity is exceeded, increasing core body temperature (Parsons, 2014; WHO, 2008). Overheating can put vital organs at

risk and leads to heat cramps, heat exhaustion, or, in the worst case, life-threatening heatstroke. At the same time, existing critical health conditions, such as chronic pulmonary and cardiac diseases, kidney disorders, and psychiatric illnesses, can be aggravated (WHO, 2015). Several studies have reported temperature-related excess mortality and increased morbidity. For example, in the summer of 2015, the second warmest summer in Switzerland on record, an increase of mortality of 5.4% was reported, with people over 75 making up 77% of the deaths (Vicedo-Cabrera et al., 2016).

Heat effects on human metabolism can also decrease labour productivity (Dasgupta et al., 2021; Sahu et al., 2013; Wyndham, 1969). Before the metabolic effects become life threatening, heat in the workplace leads to diminished physical work capacity, diminished mental ability, and increased accident risk (Kjellstrom et al., 2009c; Hancock et al., 2007). Physically demanding tasks create excess heat in the body of (outdoor) workers, rendering them particularly vulnerable to heat stress (WHO, 2008; Kjellstrom et al., 2016). Labour represents a major component of a country's national GDP and sustained losses in labour

productivity can have important repercussions on the overall economy (Carleton and Hsiang, 2016; Kjellstrom et al., 2018; Dunne et al., 2013; Kjellstrom et al., 2009a). A better knowledge of environmental heat exposure and the resulting societal impacts is of decisive importance for protecting population health and reducing economical losses.

As heat has become a major hazard in many parts of the world, heat-related mortality and labour productivity losses have become active research fields in the last years. The majority of studies focus on mortality and have assessed the burden of

past events (Vicedo-Cabrera et al., 2016; Ragettli et al., 2017; Gasparrini et al., 2015; Vicedo-Cabrera et al., 2021). But recent studies have also projected future changes in temperature-related mortality due to climate change (Martínez-Solanas et al., 2021; Gasparrini et al., 2017; Vicedo-Cabrera et al., 2016; Huber et al., 2020). On the topic of labour productivity, studies have mainly explored the potential impact of climate change from a global or continental perspective based on general circulation models (Kjellstrom et al., 2018; Casanueva et al., 2020; Orlov et al., 2019; Dasgupta et al., 2021). A few authors have also

performed studies at a national scale (Shayegh et al., 2020; Zhang and Shindell, 2021). However, so far no highly resolved projections have been done on a national scale on either of these impacts. This is however made possible as many countries are producing high spatial resolution information about the past, present and future climate for national assessments (Sørland et al., 2020). Here, we provide a detailed impact assessment and future projections for excess mortality and losses in labour productivity at the kilometre level for distinct population groups and activity classes, using a common modelling framework

(Aznar-Siguan and Bresch, 2019). This is the first work to our knowledge combining these two impact categories in the same model, which is valuable when assessing the benefits of adaptation measures.

The study is structured as follows. The data and methods for each of the two impact categories (mortality and labour productivity) are described in Section 2. Results are presented in Section 3, consisting in the impact assessment and projections





for the two issues. In Section 4, the results and the policy implications are discussed, and the main conclusions are drawn. The

data and methods for each of the two impact categories (mortality and labour productivity) are described in Section 2.

## 2    Data and Methods

We base our calculations on the risk framework proposed by the (IPCC, 2014) as implemented in the open-source CLIMADA
impact modelling platform (Aznar-Siguan and Bresch, 2019). This framework combines the three risk components *Hazard*,
*Exposure*, and *Vulnerability* to quantify socio-economic impacts of natural hazards. A schema for this modelling approach is

presented in Figure 1 for the calculation of the labour productivity losses. With this method, different impacts are explored with
the same modelling framework. We model heat-related impacts in Switzerland throughout the century, combining high resolu-
tion datasets, namely (i) the CH2018 Climate Scenarios (NCCS, 2018), national climate scenarios with daily data until 2080 on
a 2x2 kilometre grid for several RCP scenarios (*Hazard*), and (ii) the geographical distribution of population and workers per
hectare (*Exposure*). We represent the link between local exposure and heat hazards by impact functions (*Vulnerability*). The

datasets used are described in more detail in Section 2.1 and summarised in Section 1 of the SI. While the *Hazard* is spatio-
temporally variable, we assume the *Exposure* to vary only in space, and we keep the *Vulnerability* constant. We then combine
the three risk components in the modelling framework as illustrated in Section 2.2. To integrate and assess uncertainties, we
use a Monte Carlo simulation.

### 2.1    Data

We here describe the data used to model the three risk components *Hazard*, *Exposure*, and *Vulnerability*, distinguishing for
each component between data for *Mortality* and data for *Labour productivity*.

### 2.1.1    Hazard

### 2.1.2    Mortality

To compute the impact of heat on mortality, we use the daily maximum temperature ($T_{max}$) of the CH2018 Climate Sce-

narios for Switzerland. The CH2018 data were produced by applying a statistical bias-correction and downscaling method
(Quantile Mapping, QM) to the original output of EURO-CORDEX climate model simulations, using station observations and
observation-based gridded analyses as observational reference (NCCS, 2018). We choose ($T_{max}$) as an indicator as it was used
in previous analyses of heat-related mortality in Switzerland (Vicedo-Cabrera et al., 2016), including the study used to compute
the impact functions (Ragettli et al., 2017).

### 2.1.3    Labour productivity

For the impact of heat on labour productivity we use the Wet Bulb Globe Temperature (WBGT), an index that represents
apparent temperature combining several meteorological variables that regulate human heat exchanges (i.e. air temperature,




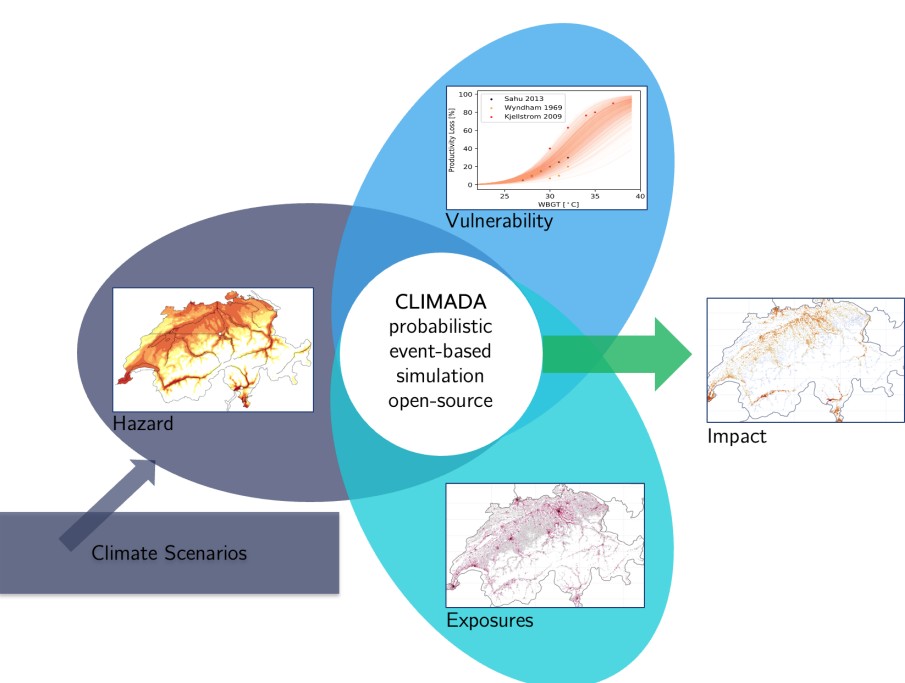

**Figure 1.** IPCC risk framework applied to the modelling of heat impact on labour productivity in CLIMADA (Aznar-Siguan and Bresch, 2019)

humidity, wind speed, and solar radiation). This metric is commonly used in research on labour productivity (Kjellstrom et al., 2018, 2009c), and also for international standards for occupational health and safety (ISO standards) (ISO, 1989). To transform
the CH2018 temperature data to hourly WBGT values, we first use daily maximum and minimum temperature data to derive hourly mean temperature values. To do so, we derive an equation predicting hourly temperature based on the daily minimum and maximum by studying daily temperature curves for the summer of 2018 at different stations in Switzerland (see Section 2 of the SI). In a second step, we estimate indoor temperatures. To understand the relationship between outdoor and indoor temperatures, we use the EnergyPlus engine to perform a thermal dynamic analysis (Roudsari and Pak, 2013). We construct
the simplified 3-D model of a typical office building in Zurich using the Rhinoceros 5 modelling tool and simulate temperature using the Honeybee plugin (Roudsari and Pak, 2013). The analysis includes the hourly energy demand of the building and the operating temperature inside. More details on the modelling can be found in Section 3 of the SI. In a final step, we transform hourly temperature values to WBGT values (see Section 4 in SI for details). We calculate hourly WBGT values for indoor and outdoor conditions, differentiating furthermore for the latter between people working under direct solar radiation and in the
shade, as sun exposure is a major component to determine the apparent temperature.





### 2.1.4 Exposure

### 2.1.5 Mortality

To calculate mortality, we use population data of the Federal Statistical Office (FSO), representing the geographical distribution of residents at hectare level (BFS, 2018). For each grid-cell, the number of residents and their age is specified. We divide the population into two age categories: people under 75 years of age and people aged 75 years and older, based on the impact functions reported by (Ragettli et al., 2017) (see Section 2.1.7).

### 2.1.6 Labour productivity

For labour productivity, we use companies' structure statistics of the FSO, depicting the geographical distribution of workers in 85 activity classes at hectare level (BFS, 2017). The dataset indicates the number of full-time equivalent employees per activity class within a hectare, which we multiply by the respective average hourly salary of each class (BFS, 2020a). Next we assign the activity classes to three main categories: low, moderate, and high physical activity (see (Kjellstrom et al., 2018)). For each activity class we furthermore specify whether the work tasks are mainly conducted inside or outside buildings. For example, we assign people working in the construction sector to the category *outside - high physical activity*, whereas people working for different types of administration to the category *inside - low physical activity* (see Section 5 of the SI).

### 2.1.7 Vulnerability

### 2.1.8 Mortality

Concerning the effect of heat on mortality, we use the impact functions for the two age categories and their uncertainty ranges from (Ragettli et al., 2017). In their work, the authors explored the relationship between temperature and daily mortality (natural causes and accidents) for 8 Swiss cities using data of the warm season (May to September) between 1995 and 2013 . The association was computed for different temperature indicators in terms of relative risk (RR) using the median temperature of the warm season (May to September) across all cities as reference. For this study we use the RR calculated for $T_{max}$. The function describes the non-linear and delayed effect of daily maximum temperature on mortality. From the mean RR estimate and the 95% confidence interval for each degree of temperature, we generate the impact functions by sampling across the confidence interval according to a normal distribution as shown in Figure 2 for the people above the age of 75 (see Section 6 of the SI for further information). We assumed the relative risk to be the same for all regions of Switzerland. This assumption is legitimated by the small size of the country and the low variability detected between the cities studied by (Ragettli et al., 2017).

### 2.1.9 Labour productivity

We generate the impact functions depicting the relationship between environmental heat exposure (i.e. WBGT values) and loss in labour productivity on the basis of 6 studies, which are listed and characterised in Table 1. We construct three impact functions and uncertainty estimations, each corresponding to a work category: low, moderate, and high physical activity (see


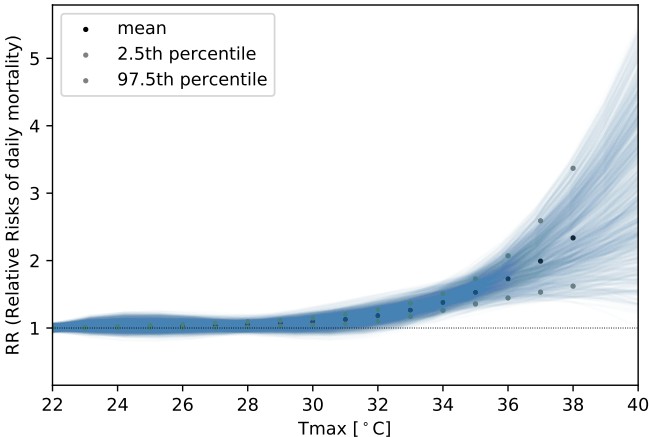

**Figure 2.** Distribution of functions describing the relationship between daily maximum temperature and relative risk of daily mortality for people with or above the age of 75. The mean curve and the confidence interval are taken from (Ragettli et al., 2017)

Section 7 of the SI). We use mainly empirical studies (i.e. studies describing the observed response of workers in different thermal environments), since they best represent the real impact of heat on labour productivity. In case there are no such studies available, we use functions derived from ISO standards, as done in previous studies (Kjellstrom et al., 2009a).

For the development of the curves, we fit the respective sigmoid functions (see Section 7.1 of the SI) using the least squares
method. Given the scarcity of studies that meet all the criteria, the large amount of uncertainty involved, and that we apply a significant amount of subjective judgement, we use a range of possible impact functions in the analysis as shown in Figure 3 for the moderate physical activity (see Section 7.2 of the SI for the impact functions of other categories). We assume a normal distribution, setting its parameters so as to take into account all the different studies when sampling the functions.

## 2.2 Methods

### 2.2.1 Mortality

First, to calculate the impact of heat on mortality, we divide the daily number of deaths (i.e. the average number of daily deaths in the summer half year between 2010 and 2019 (BFS, 2020b)) by the RR of daily mortality for each degree of temperature above 22 °C. The aim of this step is to exclude deaths attributable to heat and apply a correction factor, which enables to compute an adjusted average number of daily deaths. It is important to note that we assume that deaths are evenly distributed
throughout the warm season (May to September) and the same on each day. To calculate the daily number of deaths per cell, we multiply the computed average by the fraction of the population per cell with respect to the total Swiss population. In a further step, we calculate the fraction of deaths attributable to heat (AF) for each degree of temperature, using the equation depicted in Section 8 of the SI (Perez and Künzli, 2009). For each degree of temperature we furthermore count the number of days in the year reaching that maximum temperature in the cell. We then multiply per each degree of temperature the average

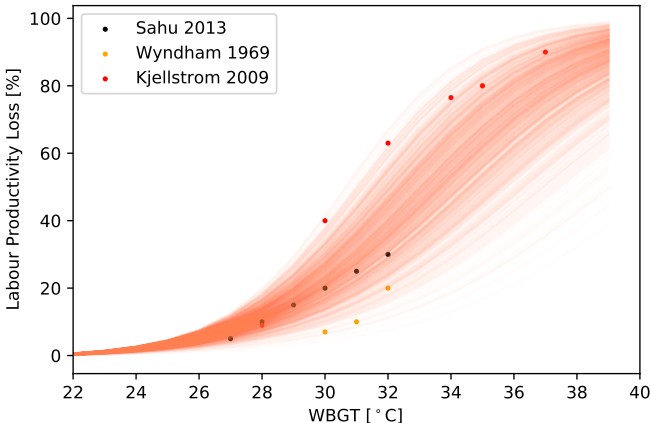

**Figure 3.** Distribution of impact functions describing the relationship between WBGT and percentage of labour productivity loss for people working at a moderate physical activity

**Table 1.** Characterization of the studies used to develop the impact functions for labour productivity. The second column illustrates for the construction of the function of which occupation category (low (*L*), moderate (*M*), and high (*H*) physical activity) the study was used. The procedure to transform results expressed as function of the temperature in WBGT (see fourth column: *Transformation to WBGT*) is illustrated in Section 4 of the SI.

| Study | Occupation category | Topic | Comments |
|---|---|---|---|
| Seppänen et al. (2006) | L | Meta-analysis on previous research on performance in office tasks at different temperatures | Temperature used as indicator |
| Zivin et al. (2018) | L | Students' performance in China | Temperature used as indicator |
| Park (2017) | L | Students' performance in the United States | Temperature used as indicator |
| Wyndham (1969) | M | Performance of mine workers on different tasks in South Africa | - |
| Sahu et al. (2013) | M/H | Performance of rice harvesters in India | - |
| Kjellstrom et al. (2009b) | M/H | Study on the impact of climate change on labour productivity | Curves are based on ISO standards |

number of daily deaths per cell, the AF, and the number of days reaching the respective temperature as maximum and sum the



obtained values up to obtain the number of heat-related deaths per year at a cell level. We then aggregate hectare values for the entire country (see modelling equation in Section 9 of the SI).

### 2.2.2 Labour productivity

We compute the impact of heat on labour productivity by multiplying the total exposed salary value at each hectare with the
hourly predicted loss in labour productivity for the respective WBGT, and summing up the values per day, throughout the year, and for the entire country (see modelling equation in Section 9 of the SI).

### 2.2.3 Uncertainty analysis

With regards to the *Hazard*, uncertainties originate from the climate simulations, the natural variability in the CH2018 climate data, and, when estimating losses of labour productivity, from the calculation of hourly WBGT. The CH2018 ensemble consists
of 12 simulations for RCP2.6, 25 for RCP4.5, and 31 for RCP8.5. Averaging those would result in losing the most extreme values, which play an essential role in this study. Therefore, we consider these as the climate model uncertainty and each time we compute the impact, we randomly pick a different simulation within the ensemble.

Because of the natural variability in the CH2018 climate data, in each run we sample uniformly a random year in a +-3 years time range. We choose this range as it proved to reduce the variability between consecutive years so that the climate trend
remains stronger.

For the transformation used to retrieve hourly temperature values from minimum and maximum daily temperature data, we assume a normal distribution (see Section 10.1 of the SI).

Also the approximation used to calculate indoor temperature values is uncertain. We assume the percentage difference for the temperature inside buildings compared to outside to have a triangular distribution and could consequently have values
anywhere from the same as outside, to 20% lower than the best estimate (see Section 10.2 of the SI).

For the relationship between temperature and WBGT values, we assume a normal distribution. We always consider the WBGT in the shadow for the indoor thermal environment, while we randomly set workers outside to work a certain amount of the day in the sun. We do this by picking a uniformly distributed number h in the range $[0, 8]$, corresponding to the number of hours that people have to work in the sun on average in a day. Then, for each hour, we randomly determine whether in that
hour people outside are working in the sun, with a probability of $h/8$.

Finally, for uncertainties related to the impact functions (*Vulnerability*) we assume normal distributions, as explained in Section 2.1.7. A summary of all uncertainties and the shape of their distributions can be found in Section 10.3 of the SI.

To integrate uncertainties in the analysis, we use a Monte Carlo approach: we calculate the impacts 1000 times where each time we randomly pick the transformations or variables that entail uncertainty from the respective distributions.




## 3    Results

Unless stated otherwise, we indicate the median estimates of the Monte Carlo simulations for specific RCP scenarios and time horizons. These represent statistical numbers, even for 2020 (the baseline), and do not represent the real number of observed deaths. It is furthermore important to note that the results refer to average years.

### 3.1    Mortality

For today's baseline, we obtain 673 heat-related deaths per year in Switzerland (see Figure 4a). 76% thereof or 518 deaths occur in the category of people over 75, albeit they only represent 10% of the total population. Under the RCP2.6 scenario, the number of death peaks in 2050 with 693 yearly heat-related deaths. According to this scenario the increase in temperature in Switzerland stops around the middle of the century. For the RCP4.5 scenario, we find 774 yearly deaths in 2050 and 861 in 2080. For the RCP8.5 scenario, the number of deaths caused by heat increases to 887 per year in 2050, exceeding 1300 in 2080. Under this scenario, the risk also highly increases for people under 75 years towards the end of the century. Uncertainties are large, for example for the RCP8.5 in 2050 the upper bound (95th percentile) is 1581 deaths, close to two times the median value - yet the lower bound (5th percentile) is at 381 deaths, as the uncertainty is not normally distributed and the median estimate is closer to the 40th percentile estimate (783 yearly deaths) than to the 60th (974 yearly deaths), as shown in Figure 4b. Detailed maps showing the spatial distribution of the impact for 2050 can be found in Section 12 of the SI.

### 3.2    Labour Productivity

With regard to the impact on labour productivity, the median estimate of labour productivity losses in Switzerland for today's baseline amounts to CHF 413 million per year (approx. $ 465 million), see Figure 5a. In 2050, the total labour productivity loss increases by only 17% in the case of the RCP2.6 scenario, but by 58% compared to today, reaching CHF 648 million per year, for the RCP8.5 scenario. The median cost for this case corresponds to about 0.17% of the total yearly exposed salary value, 0.14% for the *low physical activity* and 0.35% for the *high physical activity*. For all years and scenarios, between 50 and 70% of the losses are experienced by the low physical activity exposure type as this category actually represents 67% of the total exposure value, since most workers belong to this category and their salary is higher on average. The high physical activity type represents 20 to 35% of the losses, which means that a much higher percentage is affected, considering that only 10% of workers belong to this category. Again, the uncertainties are large, as shown in Figure 5b. For example in 2050 for the RCP8.5 the 95th percentile is of CHF 1629 million per year, which is almost three times higher than the median estimate (CHF 648 million). The 5th percentile is CHF 194 million.

Spatially explicit results for labour productivity losses of the high physical activity exposures type for the year 2050 and under the RCP8.5 scenario are shown in Figure 6. The southern cantons (Ticino, Valais, and Genève) are most impacted, as well as the area of Basel. Detailed results at cantonal level can be found in Section 13 of the SI. The relative increase in impact between the baseline and 2050 for the RCP2.6 and the RCP8.5 scenarios is illustrated in Figure 7. The relative changes in 2050 compared to the baseline are very apparent under the RCP8.5 scenario, while they are more moderate for


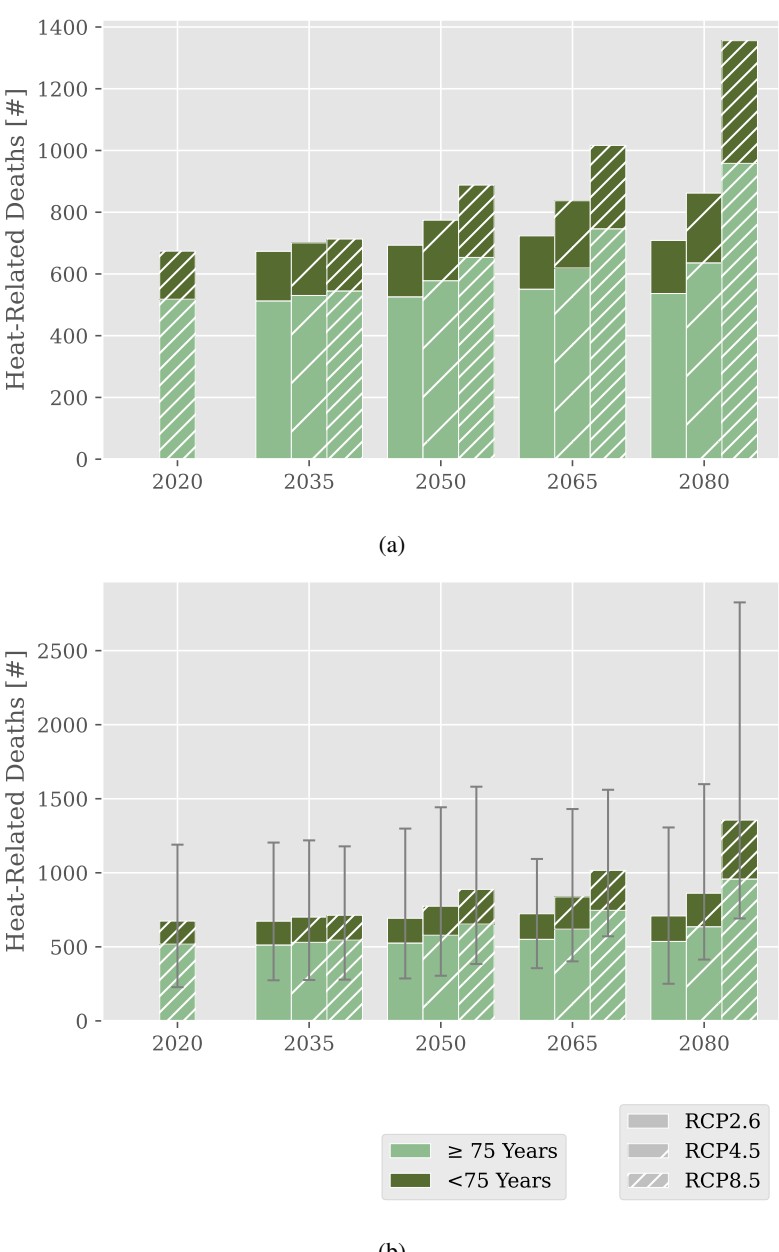

(a)

(b)

**Figure 4.** (a) Median of the number of heat-related deaths calculated in the Monte Carlo simulation for the baseline (2020, RCP8.5) and the years 2035, 2050, 2065, and 2080 under the three RCP scenarios. In (b), the uncertainty estimate corresponding to the 90% confidence interval is added





the RCP2.6 scenario. The valleys and mountainous areas experience the highest relative changes. This is because these high altitude regions currently rarely experience any days above the temperature threshold where impacts may occur, while these become more common throughout the century.

## 4 Discussion

### 4.0.1 Discussion of the results

Our results show that heat is responsible for the death of about 670 people in Switzerland and cause losses in labour productivity of CHF 413 million in an average year in today's climate.

This compares well to a recent study which estimates for the summer 2019 a heat-related excess mortality of 521 people Ragettli and Röösli (2020). For the hottest recorded summers in Switzerland in 2003 and 2015, a previous study found 975 and 804 excess deaths, respectively Vicedo-Cabrera et al. (2016). These values refer to extreme years and it is therefore expected that they would be higher than our median estimate. But these are included in the uncertainty range of our results for the present situation. However, the authors of this study only looked at summer months, while our model also takes into account heat days occurring in the spring for example.

For labour productivity, no estimations exists for past events in Switzerland. With the use of climate data, Kjellstrom et al. (2009a) predicted a total labour productivity loss of 0.1% on average over central Europe in the 2050s considering a high emission scenario. This number is very close to our estimation for Switzerland (0.17%). It is however important to note that they did consider changes in labour patterns, which is not the case in our work.

In 2080, the difference in heat impacts under a RCP8.5 scenario versus the RCP2.6 scenario are, accordingly to our model, 647 (+91%) additional heat-related deaths per year and an increase of CHF 586 millions (+140%) for labour productivity losses.

### 4.1 Uncertainties

The uncertainties are skewed towards higher values (see Figures 5b and 4b, and Section 11 of the SI for the sensitivity analysis). The climate projections used in the model NCCS (2018) are the largest source of uncertainty, stemming from natural variability and the differences between the climate simulations. For labour productivity, the second largest source of uncertainty arises from the transformation to retrieve indoor temperature values and WBGT values. Having more detailed information on the different types of buildings and being able to model them more precisely would therefore be a first key step in reducing the uncertainty. In addition, having available high resolution data for more indicators to directly compute WBGT values would also greatly increase the accuracy of the model. Finally, the modeling of the vulnerability (impact functions) appear to have a smaller contribution to the overall uncertainty. Nevertheless, the present study integrates all factors as described for the first time, and the lower bounds of our estimates proved to be robust.
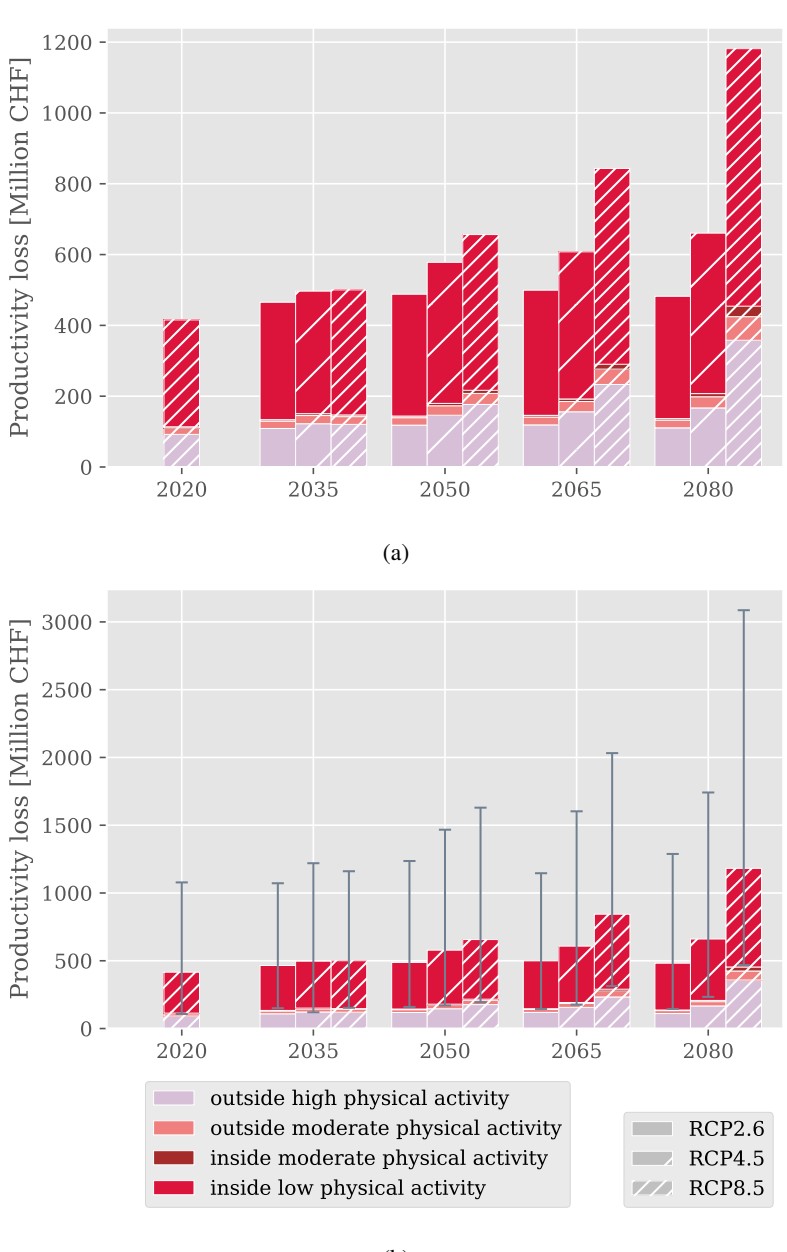

(a)

(b)

**Figure 5.** (a) Median of the labour productivity losses calculated in the Monte Carlo simulation for the baseline (2020, RCP8.5) and the years 2035, 2050, 2065, and 2080 under the three RCP scenarios. In (b), the uncertainty estimate corresponding to the 90% confidence interval is added

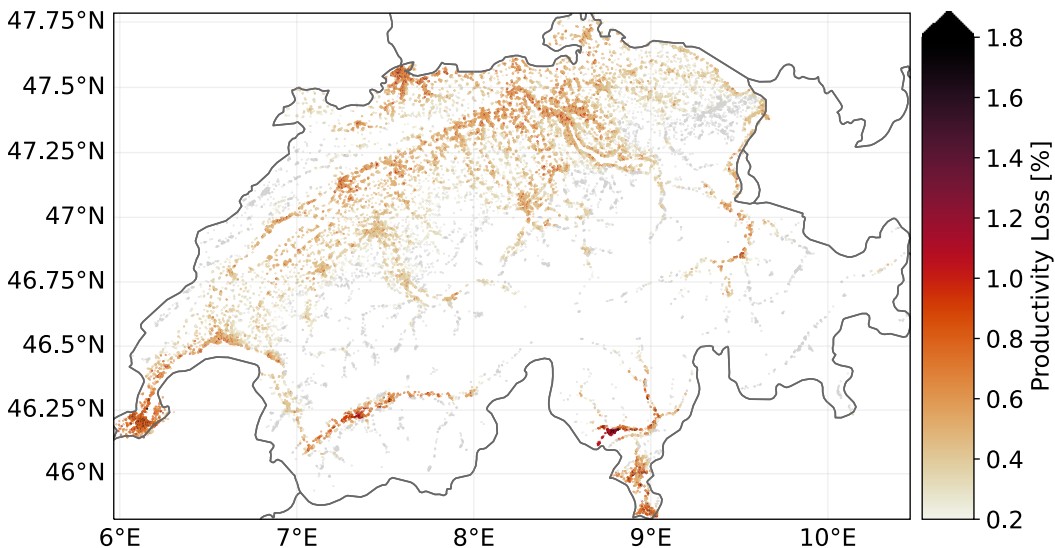

**Figure 6.** Map showing the spatial impact for labour productivity in percentage compared to the total exposure value for the year 2050, the RCP8.5 scenario, ant the category *high physical activity*

## 4.2 Limitations

A number of limitations exist in our modelling approach. The influence of heat waves, consisting of the cumulative physiological strain of consecutive hot days and the lack of rest due to hot night temperatures. Also, the CH2018 climate data does
not account for the urban heat island effect Oke (1982), hence temperatures and consequently impacts in the most densely populated areas are underestimated. Furthermore, our projections account only for the risk due to climate change and ignore demographic changes and potential effects of future adaptation of society to a warmer climate Ragettli and Röösli (2020); Martinez et al. (2019). Including these factors is an important next step to be built on the developed framework in order to fully assess the future risk. We also use one RR estimate for the whole country in the modeling of heat-related mortality. Validations
of our results are warranted using more recent and area-specific RRs. As for labour productivity, the impact functions are based on global studies and the response of workers may be different in Switzerland. We finally only consider a loss in labour productivity during working hours and do not consider a reduction in working hours due to heat, which could result in a higher loss.

## 4.3 Policy Implications

To better understand the implications of the results, these can be put in perspective with other public health issues. For example the flu provides us with an interesting comparison, as it both has an economic impact on the labour force and is responsible for a number of deaths in the population. Between 2015 and 2018, influenza caused around 600 to 700 deaths per year in Switzerland, being responsible for about CHF 100 million in direct annual healthcare costs BAG (2018) Additionally, the
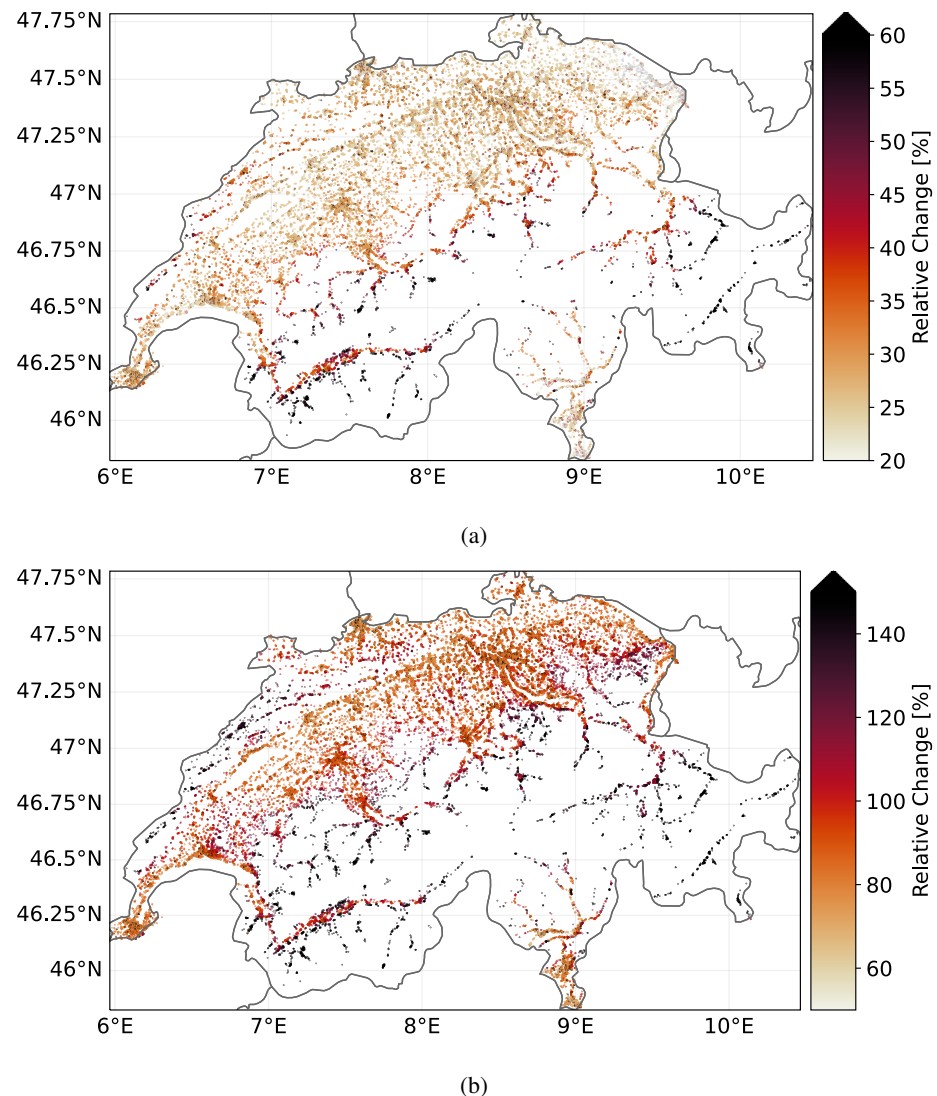

**Figure 7.** Maps showing the relative increase in the loss of labour productivity due to heat compared to the baseline for the year 2050 under the RCP2.6 (a) and the RCP8.5 scenario (b) for the *high physical activity* category





indirect economic costs due to absences from work are estimated to amount to CHF 200 million BAG (2018). Hence, already today the economic costs of heat and the number of heat-related deaths are comparable or even higher to those caused by flu.

The results regarding labour productivity costs and their increasing trend help to better understand local benefits of (global) climate mitigation progress for the Swiss economy. Our results on mortality, furthermore, indicate an increasing health burden for the Swiss population. Hence, the results of this study are valuable information for political discussions on the commensurability of additional investment costs for decarbonisation in Switzerland, allow a better understanding of the cost of inaction, and highlight the need for adaptation Bresch and Aznar-Siguan (2021).

In addition, the obtained high-resolution maps derived by the spatially explicit modelling approach are useful to identify areas that are and will be particularly affected by heat events. This reveals that, although some areas are and will remain more impacted by heat on an absolute level (for example Ticino and Genève), the relative increase due to climate change is more important for other regions (e.g. in mountainous areas). These areas may be less prepared to face high heat stress, since they have never experienced it in the past.

## 4.4 Conclusion

In this work, we use the IPCC risk framework IPCC (2014) to model the impacts of heat on mortality and labour productivity in Switzerland under changing climate, keeping socio-economic factors constant. We model the different impact categories in the same spatially explicit framework and we conduct a Monte Carlo simulation in order to quantify uncertainties. We find that the monetary costs of heat and the effects on mortality are substantial already today. The magnitude of these impacts may double (for mortality) or even triple (for labour productivity) by the end of the century should we remain on the current high greenhouse gas emissions pathway (RCP8.5). Even though uncertainties in the model are substantial, the underlying trend in impacts remains unequivocal. Future work could easily implement scenarios of dynamically changing exposure and spatially-varying vulnerability, once they become available. The model could furthermore be used to analyze adaptation measures and their effectiveness Bresch and Aznar-Siguan (2021).



*Code and data availability.*   The code and link to the input data that support the findings of this study are openly available at https://github. com/zeliest/heat_mortality_productivity_impacts.

*Author contributions.*   All authors contributed to the study design and to the manuscript. All authors read and approved the final manuscript. Z S and V N contributed equally as first authors, developing the model and writing the first draft of the manuscript. M Z and D N B contributed
to the general design of the study and the manuscript. M S R and M R contributed to the modelling of heat-related mortality and to the related parts of the manuscript. A G provided the methodology to model the temperature in building and wrote the corresponding section in the manuscript. N H was responsible for the policy applications of the model and contributed to the discussion section in the manuscript.

*Competing interests.*   The authors declare that they have no conflict of interest.

*Acknowledgements.*   We thanks Samuel Lüthi for his valuable feedback and Christopher Fairless for the language revision, as well as the
entire Weather and Climate Risks Group at ETHZ for the interest shown in our project and the provided support.





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
