# Peer review of "Projected Impact of Heat on Mortality and Labour Productivity under Climate Change in Switzerland"

_Natural Hazards and Earth System Sciences, 2021_

## Referee Comment (RC1)

Review of "Projected Impact of Heat on Mortality and Labour Productivity under Climate Change in Switzerland" by Stalhandske Z., Nesa V., Zuwald M., Ragettli M., Galimshina A., Holthausen N., Roosli M. and Bresch N.B. (Manuscript ID: NHESS-2021-1361)

Extreme heat poses a major threat on society and economy. It is therefore important to quantify the magnitude and extend of this threat in the framework of climate change. The current study addresses this need by exploiting a high-resolution climate dataset and adopting a risk assessment framework in order to investigate the impact of heat on mortality and labour productivity in Switzerland under the present and future climate conditions. The topic is very interesting and significant added value can be provided by the study. However, the paper is subject to certain major limitations in its current form.

First of all, the authors should provide a better description of the study's conceptualization and merit in the "Introduction" Section. The statements in Lines 44-48 ("A few authors [...] for national assessments.") are confusing for the reader (previous relevant studies were actually implemented at national scale, e.g., Zhang and Shindell, 2021).

The "Data and Methods" Section is too lengthy (6/15 pages of the manuscript + 14 pages in the supplementary information document) and lacks a concrete structure. It is really hard for the reader to follow the methodological concept, when he has to go through several sections and to continuously jump from the main document to the supplementary information (virtually in every paragraph in Lines 81-174). The authors should adopt a more precise and comprehensive way that will assist interpretation and will also reduce the size of this Section, including the supplementary information. In this direction, I would suggest moving Table S1 in the main document and avoiding distinguishing sub-sections based on the risk components.

Certain methods contain critical defects:

- Hourly temperature values: Which other models did you test (please provide relevant documentation)? Why did you evaluate the applied method(s) only over four stations and only for summer 2018? Which are the "few days" presented in Figure S1? Could you please provide the whole JJA model-observation timeseries over all stations used for the evaluation (Bern, Geneva, Sion and Lugano)?
- Indoor temperature values: The model used for computing indoor temperatures is based on energy and daylighting simulations (Roudsari and Park, 2013). However, it is unclear if and which weather-related data were used for the application of the model. This is very important, as outdoor weather conditions are highly associated with the indoor temperatures. More specifically, the evolution of indoor building temperatures depends not only on the energy production and consumption of a building, but also on the radiation coming through the windows and exchanged between the indoor surfaces, the natural ventilation, the generation of heat due to occupants, and the impact of the air conditioning system (Salamanca et al., 2010; Salamanca and Martilli, 2010; Matzarakis et al., 2020). The method applied by the authors neglects these critical factors.
  Further, the analysis that implemented for deriving the relationship between the indoor and outdoor temperature, lacks clarity, robustness and validity: How the *percentage difference* equation came up? Did you perform any kind of regression analysis? Why

did you apply the analysis for only three hot days in August 2018? Did you validate the proposed formula against in-situ indoor observations?

- WBGT values: It is unclear how the authors concluded to the approximation equations for computing WBGT based only on temperature. Based on Figure S2 (Why did you plot only temperatures over 20 $^{O}$C? Does this also apply for the computations?), I assume that they applied linear regression analysis between WBGT and temperature. However, it is confusing for the reader because they refer to a "model" (black line) in the legend. They also provide observations mean (red line) on the Figure for the "model" verification. However, it is not correct to validate a linear regression relationship ("model") against the same data that have been used for the development of the linear regression relationship.

  Further, the developed approximation WBGT formulas are based solely on outdoor temperature and humidity data, and a constant wind speed (Why constant? Please provide the equations used for the WBGT computations through the R package HeatStress). However, the authors apply the same formulas for the indoor environment (particularly, the shadow formula, assuming that the only difference between the indoor and outdoor environment is the lack of solar radiation in the first case, which is not true), using the estimated indoor temperatures. This cannot be considered valid. A different "model" needs to be developed and validated based on indoor measurements.

- Heat-Mortality association: How the polynomial function (equation (4) in supplementary information document) is associated with RR computed by Ragettli et al. (2017) and used in the current study? I do not agree with the assumption that the mean RR based on 8 Swiss cities is the same for all regions of the country. It may be a small country in terms of extends, but it characterized by high elevation variability, which is important for temperature and heat stress related studies. Further, the assumed low variability in RR among the 8 cities is not strongly supported by Figure S1 in Ragettli et al. (2017), e.g., Basel vs Berne.

  Mortality data are used for the period 2010 to 2019. The authors refer to them as daily deaths, but also as average daily deaths (Spatial? Temporal? It is unclear). Further, what is the geographical distribution of these data (e.g., at canton level)? What kind of deaths do you consider (e.g., all-natural)? What is the point of excluding heat attributable deaths by dividing the daily number of deaths (or average) with RR? What do you mean "evenly distributed deaths"? Why estimating the daily number of deaths per cell, when you have the actual number of deaths? In which maximum temperature do you refer in Line 143-144? Since the mortality data cover the present period (2010 – 2019), how do you estimate the number of daily deaths per cell in the future? There are so many questions raised, as the described methodology is very complicated, not adequately justified, and not easy for the reader to follow.

- Uncertainty and sensitivity analysis: Better justification needs to be provided concerning the uncertainties' distributions. Especially the assumption of a triangular distribution for the indoor temperature uncertainties is arbitrary. Further, the RMSE computed only for four stations is used in equation (9) in the supplementary information document for considering hourly values uncertainty. Also, further details are necessary concerning the sensitivity analysis process in terms of tools and methods applied and the coverage of the data used (Figure S6 refers only to 2050 under RCP8.5).

As a result of the above drawbacks in methodology, unfortunately, I believe that the outcomes of the study cannot be trusted at their present form. The authors honestly acknowledge the importance of the uncertainties and limitations in their work. However, they argue for the robustness of the study's outcomes (Lines 12-13 in Abstract, Lines 235-26 in Discussion, and Lines 272-273 in Conclusions). I am afraid that this cannot be adequately supported.

Other comments in the direction of improving the manuscript include:

- o Lines 7 and 50-51: Please clarify that the two impact categories are (i) mortality and (ii) labour productivity, as in Lines 52-55.
- o Line 63: What is the spatial resolution of the population geographical distribution used?
- o Line 65: Please clarify that SI corresponds to "Supplementary Information" document.
- o Line 106: Please replace "(see (Kjellstrom et al., 2018))" with "(Kjellstrom et al., 2018)".
- o Line 113: Please replace "(Ragettli et al., 2017)" with "Ragettli et al., (2017)". The same format change also applies to other references in the text (e.g., Line 121, Line 128).
- o Figure 2 is repeated in the supplementary information document. I would suggest replacing Figure 2 with Figure S3. The same also applies for Figure 3 and S4.
- o Mean and uncertainties in Figure 4 can be presented in a single plot. The same also applies for Figure 5.
- o The authors argue for the added value of using a high-resolution climate dataset in their analysis, but the canton level analysis is limited to two tables in the supplementary information document. I would suggest giving more details and emphasis, and promoting the regional-scale analysis.
- o Line 211: Please delete "4.0.1 Discussion of the results"
- o Line 266: Please separate "Conclusions" Section (i.e., "5. Conclusions")

References

Matzarakis, A., Laschewski, G., Muthers, S., 2020. The heat health warning system in Germany - Application and warnings for 2005 to 2019. Atmosphere (Basel). 11, 1–13. https://doi.org/10.3390/atmos11020170

Ragettli, M.S., Vicedo-Cabrera, A.M., Schindler, C., Röösli, M., 2017. Exploring the association between heat and mortality in Switzerland between 1995 and 2013. Environ. Res. 158, 703–709. https://doi.org/https://doi.org/10.1016/j.envres.2017.07.021

Roudsari, M. S. and Pak, M.: Ladybug: A parametric environmental plugin for grasshopper to help designers create an environmentallyconscious design, Proceedings of BS 2013: 13th Conference of the International Building Performance Simulation Association, 2013

Salamanca, F., Krpo, A., Martilli, A., Clappier, A., 2010. A new building energy model coupled with an urban canopy parameterization for urban climate simulations-part I. formulation, verification, and sensitivity analysis of the model. Theor. Appl. Climatol. 99, 331–344. https://doi.org/10.1007/s00704-009-0142-9

Salamanca, F., Martilli, A., 2010. A new Building Energy Model coupled with an Urban
    Canopy Parameterization for urban climate simulations-part II. Validation with one
    dimension off-line simulations. Theor. Appl. Climatol. 99, 345–356.
    https://doi.org/10.1007/s00704-009-0143-8

Zhang, Y., Shindell, D.T., 2021. Costs from labor losses due to extreme heat in the USA
    attributable to climate change. Clim. Change 164, 35. https://doi.org/10.1007/s10584-

---

## Referee Comment (RC2)

**Review of Stalhandske et al. "Projected Impact of Heat on Mortality and Labour Productivity**

**under Climate Change in Switzerland"; https://doi.org/10.5194/nhess-2021-361**

This study presents spatially explicit and aggregated projections of heat-related mortality and heat-related losses in labor productivity for Switzerland under different climate change scenarios. The topic is important given that more frequent and more intense heat events are one of the major manifestations of climate change to be expected in Central Europe. Yet, I do have a number of concerns regarding the methodology adopted, especially with regard to the estimation of heat-related mortality.

**Major:**

1) Currently, the description of the methods to compute heat-attributable mortality is difficult to follow. I would suggest moving the formulas from the supplementary material to the main text, making sure that all of the variables and parameters are properly explained.

2) To the extent I have understood the approach taken, my specific doubts concern the following

i) Why did the authors choose to model the functions coming out of Ragettli et al. (2017) as polynomial functions instead of adopting the model structure employed in Ragettli et al. (2017)? In that case, the uncertainty in the RR curves could have been assessed via the distributions of the original parameter distributions. Most importantly, there is a well-established methodological framework for computing attributable mortality based on distributed lag nonlinear models (dlnm, e.g., Gasparrini & Leone 2014, Gasparrini et al. 2015, Gasparrini et al. 2017) that the authors could have employed here.

ii) In the presentation of the current approach, it is unclear what is meant by "the fraction of the population exposed to heat" in Section 8 of the supplementary material. Using the notation of Perez & Künzli 2009, one would think that $AF_{pop} = AF_{exp}$, because all of the population should potentially be exposed to heat. Therefore, my understanding is that the formula for computing the attributable fraction per cell and $T_{max}$ value should be AF = (RR-1)/RR

iii) In Section 9 of the supplementary material, it seems like a binning of temperature values are introduced by computing the number of days within certain limits of $T_{max}$. I don't see the necessity of this. As mentioned under point (i) I would have employed the dlnm framework directly, but if you would like to start from RR curves, to my understanding you could compute the total heat-attributable mortality as

$$I_{Mortality} = \sum_{c=cells} \sum_{d=days} N_c AF_{c,d}$$

$$AF_{c,d} = \begin{cases} \frac{RR(T_{c,d})-1}{RR(T_{c,d})} & if\ T_{c,d} \geq reference\ temperature\ from\ Ragettli\ et\ al.\ 2017 \\ 0 & if\ T_{c,d} < reference\ temperature\ from\ Ragettli\ et\ al.\ 2017 \end{cases}$$

You could even consider using the cell-specific, average daily death per day of the year ($N_{c,d}$), in order to account for the seasonal structure of the baseline mortality (or for present-day the observed daily mortality). The population fractions would be used to scale the nationally available mortality data to the cell level.

3) I was also surprised to read that the authors only find a doubling of heat-related mortality by the end of the century, even under the high-emission scenario RCP8.5. Other studies, such as Gasparrini et al. 2017, Huber et al. 2020, usually find much higher rates of change for heat-related mortality under high emission-scenarios in Central Europe. Maybe this is due to the unusual approach chosen here for estimating heat-related mortality.

4) Another aspect that is currently not well explained in the method section is the definition of the baseline and future time periods. ll. 158-160 do not, e.g., mention the central years chosen. Also, it should be made explicit that "today's mortality" is based on simulated temperatures, and averaged mortality data. Given that the authors seem to have access to observed mortality data for 2010 to 2019, the question arises why the authors did not choose the 2010s as the present-day baseline, computing future impacts for selected decades in the future. If the authors decide to keep the timeslice definition adopted now, it should be justified why the 2020 baseline is only shown based on RCP8.5 data, and not for the other RCPs.

**Minor:**

l. 8: Better „We estimate that about 670 death per year are associated with heat exposure today in Switzerland." – To me "die because of heat" is quite strong, since your models are based on statistical associations

l. 22: I would suggest: "the impacts of heat on human health" rather than "metabolism". Alternatively, I would use the word "physiology" instead of "metabolism"

l. 29: Better: "people older than 75 years"

ll. 54-55: "The data and methods…" seems to be a repetition. Delete sentence?

l. 79: Would be good to include 1-2 sentences explaining the type of impact functions used.

ll. 238-9: "The influence of heat waves…": incomplete sentence

Section 4.2. Check referencing format; author names appear outside of parentheses

l. 242: references cited do not point to the pertinent literature; there are a number of studies out now including adaptive processes/demographic changes into projections of heat-related mortality under climate change (e.g., Rai et al. 2019; Lay et al. 2021, Wang et al. 2018)

ll. 202-209: You discuss here the spatially explicit results on labor productivity. It would be nice to also mention the results for mortality (currently only shown in the supplementary material) and discuss whether the areas of high risks are the same for both labor productivity and mortality.

Fig 4 and Fig. 5. To me no need to show panel (a) without error bars as differences between RCPs are visible even at larger y-axis scale.

**References:**

Gasparrini A and Leone M 2014 Attributable risk from distributed lag models. BMC medical research methodology 14 55

Gasparrini A., Y. Guo, M. Hashizume, E. Lavigne, A. Zanobetti, J. Schwartz, A. Tobias, S. Tong, J. Rocklöv, B. Forsberg, M. Leone, M. De Sario, M.L. Bell, Y.L.L. Guo, C.F. Wu, H. Kan, S.M. Yi, M. De Sousa Zanotti

Stagliorio Coelho, P.H.N. Saldiva, Y. Honda, H. Kim, B. Armstrong 2015. Mortality risk attributable to high and low ambient temperature: a multicountry observational study. Lancet, 386, pp. 369-375

Gasparrini A, Guo Y, Sera F, Vicedo-cabrera A M, Huber V, Tong S, Sousa M de, Stagliorio Z, Hilario P, Saldiva N, Lavigne E, Correa P M, Ortega N V, Kan H, Osorio S, Kyselý J, Urban A, Jaakkola J J K, Ryti N R I, Pascal M, Goodman P G, Zeka A, Michelozzi P, Scortichini M and Hashizume M 2017 Projections of temperature-related excess mortality under climate change scenarios The Lancet Planetary Health 360–7

Huber V, Krummenauer L, Peña-Ortiz C, Lange S, Gasparrini A, Vicedo-Cabrera A M, Garcia-Herrera R and Frieler K 2020 Temperature-related excess mortality in German cities at 2 ° C and higher degrees of global warming Environmental Research 186 109447

Lay C R, Sarofim M C, Vodonos Zilberg A, Mills D M, Jones R W, Schwartz J and Kinney P L 2021 City-level vulnerability to temperature-related mortality in the USA and future projections: a geographically clustered meta-regression The Lancet Planetary Health 5 e338–46

Rai M, S. Breitner, K. Wolf, A. Peters, A. Schneider, K. Chen 2019. Impact of climate and population change on temperature-related mortality burden in Bavaria, Germany. Environ. Res. Lett., 14 (2019), p. 124080

Wang Y, Nordio F, Nairn J, Zanobetti A and Schwartz J D 2018 Accounting for adaptation and intensity in projecting heat wave-related mortality Environmental Research 161 464–71

---

## Author Comment (AC1)

Answer to reviews of "Projected Impact of Heat on Mortality and Labour Productivity under Climate Change in Switzerland" by Stalhandske Z., Nesa V., Zuwald M., Ragettli M., Galimshina A., Holthausen N., Roosli M. and Bresch N.B. (Manuscript ID: NHESS- 2021-1361)

We would first like to thank the two referees for the helpful comments and suggestions to improve our work. Both reviewers expressed that our study have significant added value but that several points need to be addressed. We think that most point are relevant and will improve our study, as described in our point-by-point answer. The text written in black are the original comments provided by the referees.

In summary, both referees first thought that the methods part was hard to follow, especially when we explain the methodology regarding heat-related mortality. Based on these comments, we need to restructure this section, and include parts of the supplementary information in the manuscript. Both referees also pointed out some short-coming in the methodology regarding heat-related mortality estimations and provided helpful suggestions to improve those. One referee also pointed out that some results were unexpected, which allowed us to find a mistake in the code. We are very thankful for that and have made sure to check the rest of the code. Finally, the first referee also pointed out that the methodology to estimate the experienced temperature of workers inside and outside building should be improved. We can also answer to most of these concerns by adapting our methodology and including other data.

Review of "Projected Impact of Heat on Mortality and Labour Productivity under Climate Change in Switzerland" by Stalhandske Z., Nesa V., Zuwald M., Ragettli M., Galimshina A., Holthausen N., Roosli M. and Bresch N.B. (Manuscript ID: NHESS- 2021-1361)

Extreme heat poses a major threat on society and economy. It is therefore important to quantify the magnitude and extend of this threat in the framework of climate change. The current study addresses this need by exploiting a high-resolution climate dataset and adopting a risk assessment framework in order to investigate the impact of heat on mortality and labour productivity in Switzerland under the present and future climate conditions. The topic is very interesting and significant added value can be provided by the study. However, the paper is subject to certain major limitations in its current form.

First of all, the authors should provide a better description of the study's conceptualization and merit in the "Introduction" Section. The statements in Lines 44-48 ("A few authors [...] for national assessments.") are confusing for the reader (previous relevant studies were actually implemented at national scale, e.g., Zhang and Shindell, 2021).

We agree with the referee that these sentences are not very clear and will be reformulated. What we were trying to say is that studies have been performed previously at national level but not using such highly spatial resolved data, based on downscaled climate ensemble on a 2km grid and hectare level information on the people exposed. The dataset representing the people and workers also contain a lot of details, allowing to know the sector in which people work at a 1km resolution, as well as their age. This is very valuable information as the vulnerability highly depends on these factors. We do not know of any study on heat impacts which used this level of information.

The "Data and Methods" Section is too lengthy (6/15 pages of the manuscript + 14 pages in the supplementary information document) and lacks a concrete structure. It is really hard for the reader to follow the methodological concept, when he has to go through several sections and to continuously jump from the main document to the supplementary information (virtually in every paragraph in Lines 81-174). The authors should adopt a more precise and comprehensive way that will assist interpretation and will also reduce the size of this Section, including the supplementary information. In this direction, I would suggest moving Table S1 in the main document and avoiding distinguishing sub-sections based on the risk components.

The two referees have made comments regarding the structure and the length of the data and methods section. The modelling requires many steps, but the description needs to be simplified. We will reorganize this section by including the necessary elements for the understanding which are now in the supplementary information. The current structure, describing each component of the risk framework for both models at the same time, may not be intuitive for readers who do not work with a similar framework as we do. We therefore welcome the suggestion of the reviewer and will separate the data and methods section into the full description of the mortality model, and then the labour productivity model.

Certain methods contain critical defects:

- Hourly temperature values: Which other models did you test (please provide relevant documentation)? Why did you evaluate the applied method(s) only over four stations and only for summer 2018? Which are the "few days" presented in Figure S1? Could you please provide the whole JJA model-observation timeseries over all stations used for the evaluation (Bern, Geneva, Sion and Lugano)?

  The following two other models were tested:

  1. Method 1 from Reicosky et al. 1989.
  2. The CIBSE method from the Guide for Weather Solar and Illuminance Data, CIBSE, also described in Chow and Levermore 2007

[Figure]

*Figure 1: Observations and model for hourly temperature between for 30 days. The RMSE of the model for working hours is given.*

The timeseries for 30 days between August and July 2018 at the different stations are presented in Figure 1. We did not plot the entire time series as it becomes unreadable. This time range and station were chosen as we had access to these data already, and these are well representative for the different regions of Switzerland. Data can be obtained upon request for a larger time range and more stations if the reviewer still thinks that it is of importance in the modeling process. But we believe that on

average the error will not be significantly different and the differences in error between the stations here are negligeable as the error induced by this step is much smaller than others.

- Indoor temperature values: The model used for computing indoor temperatures is based on energy and daylighting simulations (Roudsari and Park, 2013). However, it is unclear if and which weather-related data were used for the application of the model. This is very important, as outdoor weather conditions are highly associated with the indoor temperatures. More specifically, the evolution of indoor building temperatures depends not only on the energy production and consumption of a building, but also on the radiation coming through the windows and exchanged between the indoor surfaces, the natural ventilation, the generation of heat due to occupants, and the impact of the air conditioning system (Salamanca et al., 2010; Salamanca and Martilli, 2010; Matzarakis et al., 2020). The method applied by the authors neglects these critical factors.

Further, the analysis that implemented for deriving the relationship between the indoor and outdoor temperature, lacks clarity, robustness and validity: How the percentage difference equation came up? Did you perform any kind of regression analysis? Why did you apply the analysis for only three hot days in August 2018? Did you validate the proposed formula against in-situ indoor observations?

[Figure]

*Figure 2: Daily curves for the relative difference of the inside temperature to the outside temperature for: the dashed gray line, the building model based on the 2018 hot summer days observed weather data, in blue the updated distribution used in the modelling of labour productivity losses and in red for the statistical 1/10 hottest summer for an RCP8.5 scenario for 2060. The distribution of the uncertainty was adapted to fit the future projections compared to the manuscript.*

We thank the reviewer again for the relevant comment. Indeed, it was not explained in detail in the manuscript. For the analysis of indoor temperatures we applied a dynamic hourly model, which was tuned using climate files based on .epw data. In the climate file that was used for this study, the following inputs were given: dry bulb and dew point temperatures, relative humidity, atmospheric pressure, horizontal infrared radiation intensity from sky, direct and diffuse normal and horizontal radiation, wind direction and speed, present weather observation, snow depth, and liquid precipitation depth. We do not consider air condition here. As for the relationship, we cannot test it against in-situ indoor observations. We realize that this is a simplification of what really happens in a building. But each detail cannot be addressed at this scale, especially since we consider a representative building. We however believe that this is an important contribution as most study simply assume that the temperature inside is the same as outside. The points explained in this paragraph are now carefully addressed in the manuscript.

An important improvement that we want to provide for this part, is by testing our estimation with new data that has been developed by MeteoSwiss based on the CH2018 climate scenarios specifically for building models with all needed inputs as described in the paragraph above[1]. The results of this analysis can be seen in Figure 2 for the 1 in 10-year event under an RCP8.5 for the year 2060, the red points correspond to the model results for all days where temperatures exceed 22 degrees. The distribution of the uncertainty was adapted in our model as shown in blue to a normal distribution based on the standard deviation of those points. Finally, we use the relative (or percentage) difference of indoor and outdoor temperature as we want to account for the different temperatures that can occur at a given hour. For example, a 2-degree difference would have a very different effect if it were 25 or 35 degrees outside and a relative difference can better account for the difference.

- WBGT values: It is unclear how the authors concluded to the approximation equations for computing WBGT based only on temperature. Based on Figure S2 (Why did you plot only temperatures over 20 C? Does this also apply for the computations?), I assume that they applied linear regression analysis between WBGT and temperature. However, it is confusing for the reader because they refer to a "model" (black line) in the legend. They also provide observations mean (red line) on the Figure for the "model" verification. However, it is not correct to validate a linear regression relationship ("model") against the same data that have been used for the development of the linear regression relationship.

Further, the developed approximation WBGT formulas are based solely on outdoor temperature and humidity data, and a constant wind speed (Why constant? Please provide the equations used for the WBGT computations through the R package HeatStress). However, the authors apply the same formulas for the indoor environment (particularly, the shadow formula, assuming that the only difference between the indoor and outdoor environment is the lack of solar radiation in the first case, which is not true), using the estimated indoor temperatures. This cannot be considered valid. A different "model" needs to be developed and validated based on indoor measurements.

From this comment and the one posted in the open discussion; we think that a better option for the calculation of the WBGT is to use the station level CH2018 climate projections. There are 50 stations available with all variables needed to calculate the WBGT in the HeatStress R package, and we will consider the nearest station to the working location in order to calculate the WBGT. This doesn't allow to make use of the high resolution of the gridded climate dataset but will be more scientifically sound, and the 50 stations allow for a sufficient resolution to consider the variations between the locations. Also, the exposure (people exposed) data will remain at the current resolution. We have made a request for the climate data, and we will redo the analysis using station level climate data for relative humidity, temperature, radiation, and wind speed.

As for the indoor/outdoor calculation, those were based on the R package HeatStress (Casanueva et al. 2019), which provides two WBGT calculations. One is based on Liljegren et al. 2008 and is described as "the implementation for outdoors or in the sun conditions", while the second one comes from Lemke and Kjellstrom 2012, using the formulation from Bernard et al. 1999, and is described as "corresponding to the implementation for indoors or shadow conditions". This second equation "assumes that there are no strong radiation sources (the globe temperature equals the air temperature) and wind speed of 1m/s, which corresponds to the movement of arms or legs during work." The Bernard implementation is a good approximation for the WBGT in the shadow as shown in Figure 3, but it can also be calculate using the Liljegren et al. 2008 equation, including wind speed but setting the direct
* * *
[1] https://www.meteoswiss.admin.ch/home/climate/swiss-climate-in-detail/climate-scenarios-indoor-climate.html

radiation to 0. This however does not change the estimation much, but as it can easily be
implemented, we will also do so.

[Figure]

*Figure 3: WBGT outside in the shadow calculated based on Bernard et al. 1999 and Liljegren 2008*

- Heat-Mortality association: How the polynomial function (equation (4) in supplementary
  information document) is associated with RR computed by Ragettli et al. (2017) and used in
  the current study? I do not agree with the assumption that the mean RR based on 8 Swiss
  cities is the same for all regions of the country. It may be a small country in terms of extends,
  but it characterized by high elevation variability, which is important for temperature and heat
  stress related studies. Further, the assumed low variability in RR among the 8 cities is not
  strongly supported by Figure S1 in Ragettli et al. (2017), e.g., Basel vs Berne.

[Figure]

*Figure 4: RR based on data from Schrijver at al. (2021) for 10
cantons, please note that these RR functions are based on mean
temperature, and not maximum temperature as in our study.*

The output that we get from Ragettli et al. 2017 is the RR of excess mortality at each degree
Celsius observed in the past and the 2.5[th] and 97.5[th] percentile to which we fitted a polynomial
function. The fit is only needed for the rare case where the temperature would be outside of
the past observed values (higher than 38 degrees). It is more common to use a natural cubic
spline as a fit in these kind of analysis (Vicedo-Cabrera et al. 2019), as was also done by
Ragettli et al. (2017). However, since we only look at the upper part of the function (above 22
degrees), a polynomial is a good fit.

As for the RR curves between cantons, there are significant differences and it would be more
accurate to use functions based on local observed deaths, but we did not have access to those
at the time of the analysis. A recent paper has however published an analysis looking at

difference in RR curves of heat related mortality based on mean temperature datasets (de Schrijver et al. 2021). We considered updating our study with these RR curves, from which the data fitted is shown on Figure 4. These are however do not provide the differences between the two age categories. The two main first authors of this paper are not epidemiologists and prefer to base our model on published relative risk functions. We however think that Figure 4 shows that the differences are small enough above 22 degrees for most places to be assumed to be the same over Switzerland. These are also very likely to change over time with adaptation. But ultimately, we are most interested in the changes induced by the climate and therefore also think that this assumption is reasonable.

Mortality data are used for the period 2010 to 2019. The authors refer to them as daily deaths, but also as average daily deaths (Spatial? Temporal? It is unclear). Further, what is the geographical distribution of these data (e.g., at canton level)? What kind of deaths do you consider (e.g., all-natural)?

The average (temporal) number of all cause daily deaths at canton level. We decided to use the average daily deaths in the summer months (May to September) for all of Switzerland to be consistent because the relative risk function are also swiss averages.

What is the point of excluding heat attributable deaths by dividing the daily number of deaths (or average) with RR?

The observed number of deaths already contains the heat related death. If we now add a relative risk to the number observed, this leads to an overestimation. We therefore apply a correction factor to get the average number of deaths not attributable to heat.

What do you mean "evenly distributed deaths"?

What we mean by evenly distributed is that we assume that the number of deaths is the same each day for the warm season (may to September). As seen in Figure 5, this is a reasonable assumption for these months.

[Figure]

*Figure 5: Weekly mortality in Switzerland for the years 2018 and 2013. Source:*
*https://www.bfs.admin.ch/bfs/en/home/statistics/health/state-health/mortality-causes-death.html*

Why estimating the daily number of deaths per cell, when you have the actual number of deaths?

We must estimate the number of deaths per cell (per hectar) from the total number of daily deaths in Switzerland based on how many people live in the cell.

In which maximum temperature do you refer in Line 143-144?

The maximum temperature is the daily maximum temperature.

Since the mortality data cover the present period (2010 – 2019), how do you estimate the number of daily deaths per cell in the future?

The future mortality is computed by assuming that other causes of mortality stay constant but the heat-related mortality changes based on the relative risk functions.

There are so many questions raised, as the described methodology is very complicated, not adequately justified, and not easy for the reader to follow.

- Uncertainty and sensitivity analysis: Better justification needs to be provided concerning the uncertainties' distributions. Especially the assumption of a triangular distribution for the indoor temperature uncertainties is arbitrary. Further, the RMSE computed only for four stations is used in equation (9) in the supplementary information document for considering hourly values uncertainty. Also, further details are necessary concerning the sensitivity analysis process in terms of tools and methods applied and the coverage of the data used (Figure S6 refers only to 2050 under RCP8.5).

  We agree that that the distribution for the indoor temperature is arbitrary, based on the new data that we have inputted in the building model, we decided to change it as explained above.

  For the RMSE of the hourly values, again we think that these should not be very different when considering more stations, but we can obtain more data to make sure that this is the case.

  As for the sensitivity analysis method, the model was run by blocking all variables time at a time at the maximum of the distribution except for one and running the monte-carlo simulation 1000 times. In the case of the climate model, there is no "maximum" that can be chosen, we therefore ran it for a randomly picked climate models, which is SMHI-RCA_NORESM_EUR44.

As a result of the above drawbacks in methodology, unfortunately, I believe that the outcomes of the study cannot be trusted at their present form. The authors honestly acknowledge the importance of the uncertainties and limitations in their work. However, they argue for the robustness of the study's outcomes (Lines 12-13 in Abstract, Lines 235-26 in Discussion, and Lines 272-273 in Conclusions). I am afraid that this cannot be adequately supported.

Other comments in the direction of improving the manuscript include:

We thank again the reviewer for this valuable comments. We do not address each point individually, but they will all be implemented.

o Lines 7 and 50-51: Please clarify that the two impact categories are (i) mortality and (ii) labour productivity, as in Lines 52-55.

o Line63:What is the spatial resolution of the population geographical distribution used?

The spatial resolution is of 1 hectare.

o Line65: Please clarify that SI corresponds to "Supplementary Information" document.

o Line 106: Please replace "(see (Kjellstrom et al., 2018))" with "(Kjellstrom et al.,

2018)".
o Line 113: Please replace "(Ragettli et al., 2017)" with "Ragettli et al., (2017)". The same format change also applies to other references in the text (e.g., Line 121, Line 128).

o Figure 2 is repeated in the supplementary information document. I would suggest replacing Figure 2 with Figure S3. The same also applies for Figure 3 and S4.

o Mean and uncertainties in Figure 4 can be presented in a single plot. The same also applies for Figure 5.

o The authors argue for the added value of using a high-resolution climate dataset in their analysis, but the canton level analysis is limited to two tables in the supplementary information document. I would suggest giving more details and emphasis, and promoting the regional-scale analysis.

We will additionally make the gridded hectare data available as .tiff format as soon as the results are final.

o Line211:Pleasedelete"4.0.1 Discussion of the results"
o Line266: Please separate " Conclusions" Section (i.e.,"5.Conclusions")

References

Matzarakis, A., Laschewski, G., Muthers, S., 2020. The heat health warning system in Germany - Application and warnings for 2005 to 2019. Atmosphere (Basel). 11, 1–13. https://doi.org/10.3390/atmos11020170

Ragettli, M.S., Vicedo-Cabrera, A.M., Schindler, C., Röösli, M., 2017. Exploring the association between heat and mortality in Switzerland between 1995 and 2013. Environ. Res. 158, 703–709. https://doi.org/https://doi.org/10.1016/j.envres.2017.07.021

Roudsari, M. S. and Pak, M.: Ladybug: A parametric environmental plugin for grasshopper to help designers create an environmentallyconscious design, Proceedings of BS 2013: 13th Conference of the International Building Performance Simulation Association, 2013

Salamanca, F., Krpo, A., Martilli, A., Clappier, A., 2010. A new building energy model coupled with an urban canopy parameterization for urban climate simulations-part I. formulation, verification, and sensitivity analysis of the model. Theor. Appl. Climatol. 99, 331–344. https://doi.org/10.1007/s00704-009-0142-9

Salamanca, F., Martilli, A., 2010. A new Building Energy Model coupled with an Urban Canopy Parameterization for urban climate simulations-part II. Validation with one dimension off-line simulations. Theor. Appl. Climatol. 99, 345–356. https://doi.org/10.1007/s00704-009-0143-8

Zhang, Y., Shindell, D.T., 2021. Costs from labor losses due to extreme heat in the USA attributable to climate change. Clim. Change 164, 35. https://doi.org/10.1007/s10584-

Chow, D.H.C. & Levermore, G.. (2007). New algorithm for generating hourly temperature values using daily maximum, minimum and average values from climate models. Building Services

Engineering Research & Technology - BUILD SERV ENG RES TECHNOL. 28. 237-248. 10.1177/0143624407078642.

D.C Reicosky, L.J Winkelman, J.M Baker, D.G Baker, Accuracy of hourly air temperatures calculated from daily minima and maxima, Agricultural and Forest Meteorology, Volume 46, Issue 3,1989, Pages 193-209, SSN 0168-1923

CIBSE Guide J Weather Solar and Illuminance Data, CIBSE, London, March 2002

Vicedo-Cabrera AM, Sera F, Gasparrini A. Hands-on Tutorial on a Modeling Framework for Projections of Climate Change Impacts on Health. Epidemiology. 2019 May;30(3):321-329. doi: 10.1097/EDE.0000000000000982. PMID: 30829832; PMCID: PMC6533172.

Casanueva et al. 2019. Climate projections of a multi-variate heat stress index: the role of downscaling and bias correction, Geoscientific Model Development, https://www.geosci-model-dev-discuss.net/gmd-2018-294/ Casanueva et al. 2019. Escalating environmental heat exposure – a future threat for the European workforce, Regional Environmental Change.

de Schrijver E, Folly CL, Schneider R, Royé D, Franco OH, Gasparrini A, Vicedo-Cabrera AM. A Comparative Analysis of the Temperature-Mortality Risks Using Different Weather Datasets Across Heterogeneous Regions. Geohealth. 2021 May 1;5(5):e2020GH000363. doi: 10.1029/2020GH000363. PMID: 34084982; PMCID: PMC8143899.

---

## Author Comment (AC2)

**Responses to Referee 2 for Stalhandske et al. "Projected Impact of Heat on Mortality and Labour Productivity under Climate Change in Switzerland"; https://doi.org/10.5194/nhess-2021-361**

We would first like to thank the two referees for the helpful comments and suggestions to improve our work. Both reviewers expressed that our study have significant added value but that several points need to be addressed. These points are very relevant and will improve our study, as described in our point-by-point answer. The text written in black are the original comments provided by the referees.

In summary, both referees first thought that the methods part was hard to follow, especially when we explain the methodology regarding heat-related mortality. Based on these comments, we need to restructure this section, and include parts of the supplementary information in the manuscript. Both referees also mentioned some short-coming in the methodology regarding heat-related mortality estimations and provided helpful suggestions to improve those. One referee also pointed out that some results were unexpected, which allowed us to find a mistake in the code. We are very thankful for that and have made sure to check the rest of the code. Finally, the first referee also pointed out that the methodology to estimate the experienced temperature of workers inside and outside building should be improved. We can also answer to most of these concerns by adapting our methodology and including other data.

This study presents spatially explicit and aggregated projections of heat-related mortality and heat-related losses in labor productivity for Switzerland under different climate change scenarios. The topic is important given that more frequent and more intense heat events are one of the major manifestations of climate change to be expected in Central Europe. Yet, I do have a number of concerns regarding the methodology adopted, especially with regard to the estimation of heat-related mortality.

**Major:**

1. 1)  Currently, the description of the methods to compute heat-attributable mortality is difficult to follow. I would suggest moving the formulas from the supplementary material to the main text, making sure that all of the variables and parameters are properly explained.

   The other referee also indicated that this part of the methodology was hard to follow. We agree that it will be easier to follow if we include the formulas an explain the steps based on each variable of the formulas.

2. 2)  To the extent I have understood the approach taken, my specific doubts concern the following
   i)  Why did the authors choose to model the functions coming out of Ragettli et al. (2017) as polynomial functions instead of adopting the model structure employed in Ragettli et al. (2017)? In that case, the uncertainty in the RR curves could have been assessed via the distributions of the original parameter distributions. Most importantly, there is a well-established methodological framework for computing attributable mortality based on distributed lag nonlinear models (dlnm, e.g., Gasparrini & Leone 2014, Gasparrini et al. 2015, Gasparrini et al. 2017) that the authors could have employed here.

   Ragettli et al. (2017) follows the distributed lag nonlinear models methodology as done in the cited publications. We employ the resulting RR for the lag following the heat days from this study. This is the same methodology as adopted by Gasparrini et al. (2017) for example. As for the polynomial function, this was adopted instead of the commonly used cubic splines functions, as we only look at the rising part of the function above 22 degrees.

ii) In the presentation of the current approach, it is unclear what is meant by "the fraction of the population exposed to heat" in Section 8 of the supplementary material. Using the notation of Perez & Künzli 2009, one would think that $AF_{pop}$ = $AF_{exp}$, because all of the population should potentially be exposed to heat. Therefore, my understanding is that the formula for computing the attributable fraction per cell and $T_{max}$ value should be AF = (RR- 1)/RR

We thank the reviewer for the relevant comment. This is indeed the case; we do consider the fraction of the population exposed as being the full population per cell o the corresponding age category. We copied the formula given by Perez & Künzli (2009), but we should specify that in our case it is simplified to AF = (RR- 1)/RR. We agree that this is confusing and will explain the simplification of the equation.

iii) In Section 9 of the supplementary material, it seems like a binning of temperature values are introduced by computing the number of days within certain limits of $T_{max}$. I don't see the necessity of this. As mentioned under point (i) I would have employed the dlnm framework directly, but if you would like to start from RR curves, to my understanding you could compute the total heat-attributable mortality as

$$I_{Mortality} = \sum_{c=cells} \sum_{d=days} N_c AF_{c,d}$$

$$AF_{c,d} = \begin{cases} \frac{RR(T_{c,d})-1}{RR(T_{c,d})} & if\ T_{c,d} \geq reference\ temperature\ from\ Ragettli\ et\ al.\ 2017 \\ 0 & if\ T_{c,d} < reference\ temperature\ from\ Ragettli\ et\ al.\ 2017 \end{cases}$$

This is the same equation as we are using, but it is again a sign that our methodology was not clear and that it must be better explained, by also including the equation in the methods part.

You could even consider using the cell-specific, average daily death per day of the year ($N_{c,d}$), in order to account for the seasonal structure of the baseline mortality (or for present-day the observed daily mortality). The population fractions would be used to scale the nationally available mortality data to the cell level.

[Figure]

*Figure 1: Weekly mortality in Switzerland for the years 2018 and 2013. Source:*
*https://www.bfs.admin.ch/bfs/en/home/statistics/health/state-health/mortality-causes-death.html*

This is also something that we tried to implement; we were however limited by computational power as it doesn't allow to sum all days of same temperature. The next best estimation is to considerer the average daily deaths in summer months, which does not vary much as shown in Figure 1.

3.  I was also surprised to read that the authors only find a doubling of heat-related mortality by the end of the century, even under the high-emission scenario RCP8.5. Other studies, such as Gasparrini et al. 2017, Huber et al. 2020, usually find much higher rates of change for heat-related mortality under high emission-scenarios in Central Europe. Maybe this is due to the unusual approach chosen here for estimating heat-related mortality.

    We are extremely thankful for this comment. Upon checking the referenced papers, we realized that indeed our numbers were out of the range that we should expect. As explained above, the methodology remains very similar although it is slightly simplified, and we would expect similar results. it also made us realize that we were predicting a larger increase for the mortality of people in the category of people under 75 compared to those over, which should not be the case. Upon checking the code, we found that we exchange the RR curves for the two categories. This was corrected, and the new results that we would get for mortality are shown in Figure 2.

    Huber et al. (2020) report a 2.8 average factor of change in mortality due to heat under a global 3 degree increase in temperature for Germany, which is equivalent to an RCP8.5 scenario in 2080, for which we estimate a change factor of about 2.6 In the study by Gasparini

[Figure]

*Figure 2: Updated results for heat related mortality, as we ran only 100 simulations, these may still slightly change.*

    et al. (2017), they project about a 4 time increase since 2020. We think that this difference may be explained by the fact that the authors look at all central Europe. Also, we can expect differences coming from the climate models, as we see an important variability between them for our model. The CH2018 RCP8.5 data are based on 32 simulations of 15 global and regional climate models with different initial conditions and two resolution, while the model by Gasparini et al. (2017) and Huber et al. (2020) are run on 5 global climate models. We will however make sure to compare our results in the discussion with previous literature.

4.  Another aspect that is currently not well explained in the method section is the definition of the baseline and future time periods. ll. 158-160 do not, e.g., mention the central years chosen. Also, it should be made explicit that "today's mortality" is based on simulated temperatures, and averaged mortality data. Given that the authors seem to have access to

observed mortality data for 2010 to 2019, the question arises why the authors did not choose the 2010s as the present-day baseline, computing future impacts for selected decades in the future. If the authors decide to keep the timeslice definition adopted now, it should be justified why the 2020 baseline is only shown based on RCP8.5 data, and not for the other RCPs.

We think that it is important to also show what we model for today's climate to see the change resulting from the climate signal. We will however make this clearer in the manuscript. As for the observed excess mortality, we prefer to provide it in the text as it becomes too much information in the same plot.

**Minor:**

We agree with all following comments and are happy to implement those.

l. 8: Better „We estimate that about 670 death per year are associated with heat exposure today in Switzerland." – To me "die because of heat" is quite strong, since your models are based on statistical associations

l. 22: I would suggest: "the impacts of heat on human health" rather than "metabolism". Alternatively, I would use the word "physiology" instead of "metabolism"

l. 29: Better: "people older than 75 years"
ll. 54-55: "The data and methods..." seems to be a repetition. Delete sentence?
l. 79: Would be good to include 1-2 sentences explaining the type of impact functions used. ll. 238-9: "The influence of heat waves...": incomplete sentence
Section 4.2. Check referencing format; author names appear outside of parentheses

l. 242: references cited do not point to the pertinent literature; there are a number of studies out now including adaptive processes/demographic changes into projections of heat-related mortality under climate change (e.g., Rai et al. 2019; Lay et al. 2021, Wang et al. 2018)

ll. 202-209: You discuss here the spatially explicit results on labor productivity. It would be nice to also mention the results for mortality (currently only shown in the supplementary material) and discuss whether the areas of high risks are the same for both labor productivity and mortality.

Fig 4 and Fig. 5. To me no need to show panel (a) without error bars as differences between RCPs are visible even at larger y-axis scale.

**References:**

Gasparrini A and Leone M 2014 Attributable risk from distributed lag models. BMC medical research methodology 14 55

Gasparrini A., Y. Guo, M. Hashizume, E. Lavigne, A. Zanobetti, J. Schwartz, A. Tobias, S. Tong, J. Rocklöv, B. Forsberg, M. Leone, M. De Sario, M.L. Bell, Y.L.L. Guo, C.F. Wu, H. Kan, S.M. Yi, M. De Sousa Zanotti

Stagliorio Coelho, P.H.N. Saldiva, Y. Honda, H. Kim, B. Armstrong 2015. Mortality risk attributable to high and low ambient temperature: a multicountry observational study. Lancet, 386, pp. 369-375

Gasparrini A, Guo Y, Sera F, Vicedo-cabrera A M, Huber V, Tong S, Sousa M de, Stagliorio Z, Hilario P, Saldiva N, Lavigne E, Correa P M, Ortega N V, Kan H, Osorio S, Kyselý J, Urban A, Jaakkola J J K, Ryti N R I, Pascal M, Goodman P G, Zeka A, Michelozzi P, Scortichini M and Hashizume M 2017 Projections of temperature-related excess mortality under climate change scenarios The Lancet Planetary Health 360– 7

Huber V, Krummenauer L, Peña-Ortiz C, Lange S, Gasparrini A, Vicedo-Cabrera A M, Garcia-Herrera R and Frieler K 2020 Temperature-related excess mortality in German cities at 2 ° C and higher degrees of global warming Environmental Research 186 109447

Lay C R, Sarofim M C, Vodonos Zilberg A, Mills D M, Jones R W, Schwartz J and Kinney P L 2021 City-level vulnerability to temperature-related mortality in the USA and future projections: a geographically clustered meta-regression The Lancet Planetary Health 5 e338–46

Rai M, S. Breitner, K. Wolf, A. Peters, A. Schneider, K. Chen 2019. Impact of climate and population change on temperature-related mortality burden in Bavaria, Germany. Environ. Res. Lett., 14 (2019), p. 124080

Wang Y, Nordio F, Nairn J, Zanobetti A and Schwartz J D 2018 Accounting for adaptation and intensity in projecting heat wave-related mortality Environmental Research 161 464–71

---

## Referee Report (RR1)

Review of "Projected Impact of Heat on Mortality and Labour Productivity under Climate Change in Switzerland" by Stalhandske Z., Nesa V., Zuwald M., Ragettli M., Galimshina A., Holthausen N., Roosli M. and Bresch N.B. (Manuscript ID: NHESS-2021-1361) **REVISION 1**

I would like to thank the authors for addressing all the review comments. The manuscript is improved and justified. Thus, I suggest the paper to be accepted for publication in NHESS.

---

## Author Response (AR2)

Point by point response:

The revised manuscript has considerably improved compared to the first version. Descriptions and discussions are now much easier to follow. Overall, it is nice to see a comprehensive assessment of heat stress on health and labor productivity under different scenarios of climate change for a European country.

I think that the manuscript is now close to a version that can be published. I only have two more reservations regarding the computation of heat-related mortality.

Since the framework of attributable fraction (computed as AF=(RR-1)/RR) is used, I was surprised to read that the authors scale the observed averaged daily mortality by RR before multiplying it with AF to derive attributable mortality (ll. 88-90). To my understanding the authors should simply use the total observed averaged daily mortality.

The heat-attributed deaths are already considered in the observed deaths, if we add on top the attributable fraction at a given temperature, we would be overestimating the heat-attributed deaths. We have added an explanation to that point.

The other issue that is still not really clear to me is why the authors do not use a continuous temperature scale, but consider the RR only per full degree of Tmax. I assume that they simply round the maximum temperature to the nearest integer, thereby actually binning the temperature into Tmax+/-0.5°C. Unless there are computational restrictions or issues of data availability, I would think that the more straightforward approach would be to compute the RR on a finer scale, by simply including the Tmax per day at the resolution available (if >22°C) into the polynomial function given in the appendix, and finally summing attributable deaths over all days.

We did indeed use only full degree of Tmax, this is because the data provided from the previous study are only given for these temperatures. We also don't think that this would lead to substantial changes in the results.

If the authors stick to the approach that is currently used I think these two issues still need further explanation and justification.

In addition, I have the following minor points:
- l.160: "the real number of observed deaths" is a bit misleading. Maybe write something like "and use the averaged observed daily deaths as the mortality baseline"
- l.225: "The influence of heat waves…" is an incomplete sentence.

Thank you for both comments, these have been implemented

Last but not least, merely as an advice for future work, I would like to point out that it has been a bit difficult to see the changes made to the manuscript because the tracked changes version did only include the major corrections. Also, the authors could have included line numbers in their responses to guide the reviewers to the sentences/parts where changes were introduced, even if they were minor.

Thank you for the comment, this was due to the text being written in Latex but we have now found how to include track changes in the pdf generated from latex.

Additional changes:

We have corrected a few formulations from the text and added two points to the limitations which we have realized are important to mention based on recent work.